# Ritualized aggressive behavior reveals distinct social structures in native and introduced range tawny crazy ants

Edward G. LeBrun[1]*, Robert M. Plowes[1], Patricia J. Folgarait[2], Martin Bollazzi[3], Lawrence E. Gilbert[4]

**1** Brackenridge Field Laboratory, Department of Integrative Biology, The University of Texas at Austin, Austin, Texas, United States of America, **2** Programa de Investigación en Interacciones Biológicas, Centro de Estudios e Investigaciones, Universidad Nacional de Quilmes, Buenos Aires, Argentina, **3** Departamento de Protección Vegetal, Unidad de Entomología, Facultad de Agronomía, Universidad de la Republica, Montevideo, Uruguay, **4** Department of Integrative Biology, The University of Texas at Austin, Austin, Texas, United States of America

\* edwardlebrun@austin.utexas.edu

**Data Availability Statement:** All relevant data are within the paper and its Supporting Information files.

## Abstract

How workers within an ant colony perceive and enforce colony boundaries is a defining biological feature of an ant species. Ants fall along a spectrum of social organizations ranging from single-queen, single nest societies to species with multi-queen societies in which workers exhibit colony-specific, altruistic behaviors towards non-nestmate workers from distant locations. Defining where an ant species falls along this spectrum is critical for understanding its basic ecology. Herein we quantify queen numbers, describe intraspecific aggression, and characterize the distribution of colony sizes for tawny crazy ant (*Nylanderia fulva*) populations in native range areas in South America as well as in their introduced range in the Southeastern United States. In both ranges, multi-queen nests are common. In the introduced range, aggressive behaviors are absent at all spatial scales tested, indicating that within the population in the Southeastern United States *N. fulva* is unicolonial. However, this contrasts strongly with intraspecific aggression in its South American native range. In the native range, intraspecific aggression between ants from different nests is common and ritualized. Aggression is typically one-sided and follows a stereotyped sequence of escalating behaviors that stops before actual fighting occurs. Spatial patterns of non-aggressive nest aggregation and the transitivity of non-aggressive interactions demonstrate that results of neutral arena assays usefully delineate colony boundaries. In the native range, both the spatial extent of colonies and the average number of queens encountered per nest differ between sites. This intercontinental comparison presents the first description of intraspecific aggressive behavior for this invasive ant and characterizes the variation in colony organization in the native-range, a pre-requisite to a full understanding of the origins of unicoloniality in its introduced range.

**Funding:** This work was supported by the Lee and Ramona Bass Foundation to LEG, and RP and a grant from the University of Texas International Office to LEG. The funders had no role in study design, data collection, analysis, decision to publish, or preparation of the manuscript.

**Competing interests:** The authors have declared that no competing interests exist.

## Introduction

How social insect colonies define the boundaries of their societies is fundamental to their biology. The process of social discrimination, the identification and acceptance of colony members and rejection of aliens, occupies an analogous role in the biology of the colonial super-organism as self-, nonself-recognition of tissues does at the organismal level. Ant colonies fall along a spectrum of social organization ranging from single-nest, single-queen colonies antagonistic to all others to multi-nest, multi-queen colonies varying widely in size, reaching, in the extreme, unicolonial populations where workers treat non-nest-mate workers as colony members over any geographic distance [1,2]. Defining where a species falls along this spectrum and quantifying the variation that colonies exhibit in this trait is critical for understanding the basic ecology of a species. Defining these traits for an invasive ant species takes on additional importance and complexity as a complete picture requires characterizing the nature of intraspecific aggression and using that knowledge to define the limits and organization of colonies in both the introduced and native range of that species.

*Nylanderia fulva* (Mayr) (tawny crazy ants) are an invasive ant of emerging importance. Originally from southern South America, these ants were introduced into Colombia sometime prior to 1971 and have been documented causing substantial ecological and agricultural impacts in mid-elevation environments [3–5]. In North America, dense local-populations of this ant were first encountered in Florida in 1990's and the eastern Gulf region of Texas and in the early 2000's [6–8]). Complicating this more recent invasion history, the taxonomic identity of this invader in North America was unresolved until 2012 and has since undergone generic revision, so it has been referred to variously as *Paratrechina* sp. nr. *pubens*, *Nylanderia* sp. nr. *pubens*, and Rasberry crazy ants [9]. Through human-mediated, jump dispersal local-populations of *N. fulva* have now established in every Gulf coast state and in counties throughout Florida and the Gulf region of Texas [10]. Both red imported fire ants and Argentine ants originate in South America and have spread from established populations in the Southeastern United States to global distributions [11–13]. With their current North American bridgehead, tawny crazy ants are poised to follow that same path to introduction into other parts of the world.

In their introduced range, *N. fulva* colonies are highly polygynous, and polydomous and occupy a variety of nest sites varying from soil cavities, rotten logs, and leaf litter to electrical-switch boxes and trash [3]. Introduced, local-populations can reach extreme densities, and are known to reduce the abundance and species richness of native ants, other arthropods, and negatively impact some reptiles [4,14]. Female reproductives are not known to engage in mating flights although males do fly, thus mating is presumed to be intranidal [5]. Following introduction, colonies spread through nest fission, leading to a contiguous area of habitat occupied by a network of interconnected nests. Within introduced local-populations, ants from different nests do not interact aggressively, so these local-populations constitute all or part of spatially-expansive supercolonies [15,16].

Part of the reason that invasive ants like *N. fulva* are thought to be both successful invaders and ecologically damaging lies in this supercolonial social organization [17]. Most ecologically damaging, introduced ant species tend to exhibit a supercolonial social structure in their introduced ranges [18]. However, few of these species have had their social organization described in both native and introduced populations [1,19–21]. Among these investigations, evaluating differences in population genetic structure have been prioritized over understanding the behaviors that functionally determine the permeability of boundaries between colonies. Thus, existing intercontinental comparisons of social organization remain limited in species coverage and scope.

Social organization arises from the behavioral responses to colony identity signaling in ants. Colony identity signaling is mediated by recognition of shared surface chemical profiles. These profiles are both genetically and environmentally determined and typically comprised of cuticular hydrocarbons [22]. The behaviors elicited by contact with the chemical profiles of colony-mate or foreign workers functionally determine colony boundaries [23,24]. The intraspecific aggressive behaviors triggered by contact with foreign workers vary considerably among ant species [25–28] and interaction context [29–31]. In the case of *N. fulva*, in North America intraspecific aggression in a neutral arena context has been noted to be absent both within [32] and between [15] local populations. However, before this absence of aggressive behavior can be meaningful interpreted, we must first know the behaviors the focal species uses to express intraspecific aggression and whether these behaviors are expressed in the neutral arena context being considered.

Further to understand the origins of the apparent unicoloniality of *N. fulva* in the introduced range [15], we must first understand the native state from which this organization is derived. What constitutes intraspecific aggressive behavior in *N. fulva*? Are there behavioral boundaries between colonies in the native range? Does *N. fulva* form polydomous (multinest), polygyne (multiqueen), spatially-expansive colonies in its native range? And finally, how do colony extent and queen number vary in its native context? Answers to these questions will provide the essential context for understanding the developmental or evolutionary trajectory that this ecologically important trait has undergone post introduction as well as enhancing understanding of social organization transitions in ant invasions generally. We employ an intercontinental comparative framework to characterize the behaviors associated with intraspecific aggression in tawny crazy ants and use this information to describe the organization and extent of colonies of this ant in its native and introduced range in North America.

## Methods

### Nest collection

Ant nests were collected in Corrientes province, Argentina in 2015 and in Uruguay in 2016 (the native range), in Texas and Florida in 2016 and 2017 (the introduced range) (Fig 1). Nests were located by turning rocks and opening rotting logs. All visible ants, the top 1 cm of soil, and nest material (rotting wood) containing ants were transferred to a large plastic tray. *N. fulva* nests superficially, so this approach captures a large fraction of the ants in a nest. However, ants in cavities deeper in the soil were missed by this collection method. The number of queens found in a nest thus represents a representative sample not a complete collection. For nests in the native range and for nests in the introduced range used in aggression assays, ants were then hand separated from nest material using aspirators and paint brushes to brush ants off surfaces. From these separated ants, all queens, males, and winged gynes were removed and preserved in 100% ethanol. Using chambered aspirators, up to 300 live ants were then separated into 5 x 10 x 8 cm nest box containers with a moist paper towel as a nest site and a cotton ball soaked with 30% sugar water for food. The top two cm of the nest box walls were coated with Fluon™ to prevent ant escape. Ants were used for interaction assays within a maximum of 10 days of collection, with most interaction assays conducted within 3 days (238 out of 270 nest pairs).

In the introduced range, nest collections to assess queen numbers per nest were done separately from nest collections made for worker aggression assays. Nest collections assessing queen numbers were made in 2017 and 2018. These nests were collected in a similar manner as above, but ants were not separated from nesting material by hand. Instead, using a heat lamp to warm and dry nest material, ants were forced to relocate to cool, moist refuges:

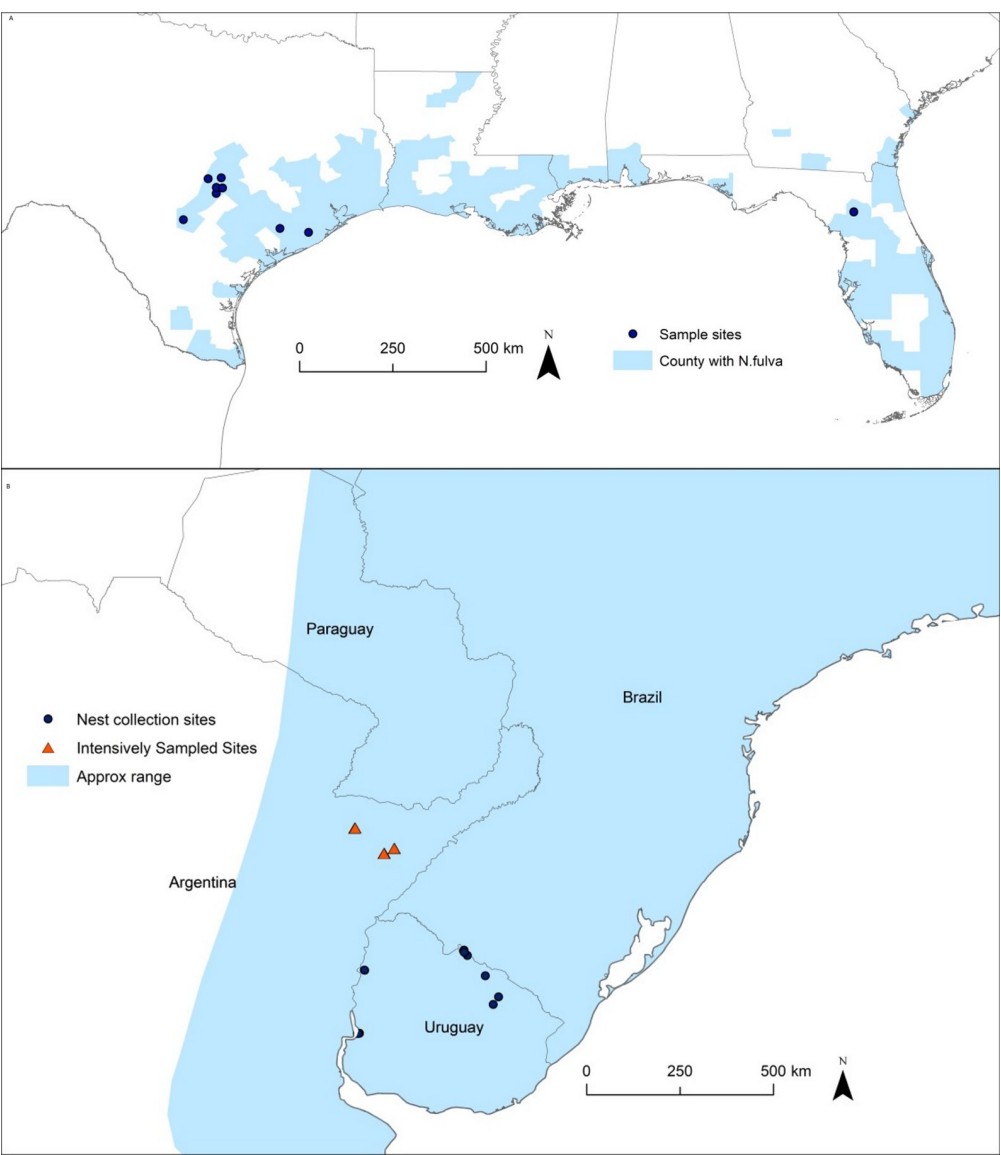

**Fig 1. Map of study sites.** (A) North America. Areas in blue are counties with at least one known *N. fulva* infestation. (B) South America. Areas in blue are part of the native range [33]. GPS co-ordinates provided in S1 Table.

cotton-plugged test tubes half filled with water. For aggression assays, *N. fulva* nests were collected from 10 widely scattered sites in Texas and an additional site in Florida (S1 Table). From each site, ants were collected from two nests separated by 100 m. All sites were in distinct local-populations: defined as the contiguous area occupied by *N. fulva* surrounding a point of introduction. In most local-populations, *N. fulva* nests were typically dense and ubiquitous throughout this area. Introduced-range sites varied from native woodlands with little active management, to heavily grazed pastures, to urban parkland and creek drainages.

In Argentina, three sites in the province of Corrientes were intensively searched for nests to reveal behavioral boundaries and seek to define colony identities (S1 Table). At these sites, within the search zone, all nests encountered were sampled if they were more than 3 m from the nearest, previously sampled nest. Site 1 was a small town located within a reserve. Forty-six nests were collected along the residential road shoulders throughout the town. The town

consisted of a grid of dirt roads with blocks made up of a mixture of wood lots, pasture, and small residences. *N. fulva* nests were scattered, common in some areas of town and rare in others. Site 2 was on a private reserve. Nineteen nests were encountered along a belt of habitat 1.7 km long. Habitat was a mixture of open grassland areas and copses of dense, diverse woodlands ("monte"). Site 3 was in a national park. The habitat was flood-prone open grasslands with scattered forested uplands. Ten nests were encountered in a transect running along both shoulders of the main park road. *N. fulva* was generally scarce in the park and found only on a 1.2 km section of the road near the park entrance (Fig 1). The surrounding grassland was flooded at the time of sampling.

In Uruguay, to examine the frequency of spatially-expansive colonies, *N. fulva* nests were collected at 11 sites separated by at least 1 km and up to 170 km (S1 Table). All sites were located adjacent to roadsides, frequently where the road crossed a drainage feature. Sites were selected areas where non-plantation woodlands intersected pastureland. All sites were heavily disturbed, many were periurban and many were along the edges of flood plains. Two to three nests separated by a minimum of 40 m were collected per site (Fig 1).

## Neutral arena aggression assays

To characterize intraspecific aggressive behaviors and to assess the colony membership of a nest, we conducted neutral arena interaction assays. Aggression assays consisted of conflicts between 5 workers from each nest staged in neutral arenas (5 x 5 neutral arena assays). This type of neutral arena assay has been found to yield repeatable results in other systems [29]. Neutral arenas were 20 ml clear glass scintillation vials, 5.5 X 2.5 cm (height x diameter), the walls of which coated with Fluon™ excepting the bottom centimeter left clear for observation. The following criteria and procedures for the aggression assays were developed over several initial trials until repeatable outcomes were achieved. Five workers were removed from each nest fragment by aspiration and deposited into separate vials. Vials were maintained at 30˚ C. Ants in both vials were given at least 10 minutes to adjust to their surroundings before being combined into a single vial by tipping one vial into the other. Starting one minutes afterwards, arenas were observed for 20 seconds every two minutes for a total of four observations. Vials were cleaned with ethanol between uses. Analysis of aggression in nest-pairs run both during the day and after dark indicated reduced aggression at night. All aggression assays included herein were run during daylight hours (for protocol see: dx.doi.org/10.17504/protocols.io. 8z9hx96).

In the native range, interaction assays were typically repeated three times per nest-pair. However, replication of nest-pairs varied from one, when insufficient ants were available, (15 out of 156 pairs) to 14 times. Replication values higher than three resulted from including nest pairs known to be aggressive or non-aggressive as internal controls for assay conditions within sets of neutral arena assays. In the introduced range, where little variation in aggression was found, arenas and nest-pairs were replicated twice. In each successive run of a nest-pair, the order of introduction (e.g. workers from nest A introduced into the vial containing workers from nest B) was reversed. In all assays, the observer was blind to the identity of the nests being combined.

To compare the dependency of aggression on nest separation distance between the native and introduced ranges, interactions were divided into four categories based on separation distance: 'nestmate', 'close', 'distant', and 'different-region'. 'Nestmate' interactions were comprised of ants that were collected from the same nest into separate nest boxes, held apart for at least 24 hours before being combined in a neutral arena. These provide a control measure of nest-mate interaction scores. 'Close' interactions were ants collected from different nests

within the same site. In local-populations in the introduced-range, nests are dense and easily collected, and 'close' nest-pairs were separated by 60–100 m. In the native-range, local-populations are sparser, and nest collection necessarily more opportunistic. There 'close' nest-pairs were separated by 10–100 m. 'Distant' interactions were between ants collected from disjunct areas. In the introduced range, sites containing *N. fulva* are distributed as disjunct local-populations surrounding independent points of introduction; all 'distant', introduced-range nest-pairs were collected from different local-populations (separation distance: 117 ± 95 km) (mean ± SD). In the native-range, *N. fulva* distributions are more continuous. Native-range, 'distant' interactions were defined as pairings of nests separated by more than 1 km. Most native-range, 'distant' nest-pairings were separated by intervening habitat types and much larger distances (62 ± 11 km). 'Different-region' interactions occurred only in the introduced range and denoted ants collected in Texas and Florida (1449 ± 81 km). In the native range, regions (Argentina and Uruguay) were sampled in different years, so 'different-region' nest-pair confrontations were not possible in the native range.

## Aggression scale

Standard aggression scales have been developed to quantify intraspecific interactions in Argentine ants (*Linepithema humile*) [34,35]. We conducted preliminary interaction assays to adapt this scale to incorporate the behaviors observed in interactions between *N. fulva* workers. Data from these preliminary assays are not included in the analyses presented herein. This *N. fulva* specific aggression scale takes values from 0 to 5 (Table 1). Scores of 0 arise from brief, non-aggressive contact. Scores of 1 are non-aggressive and typical for nest-mate or colony-mate interactions. Scores of 2 code for very brief biting interactions, and, because they are brief, they can be difficult to interpret potentially reflecting low-level aggression or a brief bout of allogrooming between colony-mates. Values of 3 and above denote clearly aggressive behaviors. Scores of 3 occurred when the aggressor bites and holds its opponent for more than 3 seconds. Scores of 4 occurred when the aggressor while biting and holding its opponent pointed its acidopore at the aggressee without spraying venom. *N. fulva* venom, like all formicine ants, is formic acid. They deploy it by rearing up on their hind and middle legs, curling their gaster under their body, pointing the tip at their opponent and spraying. Scores of 5, the highest level of aggression, occurred when either both ants grappled with their mandibles, forming a ball and fighting or the aggressor sprayed formic acid at its opponent. Table 1 provides definitions for all behaviors. The Results section describes unusual behaviors observed in this system. Because aggression typically escalates, for our measure of aggression we averaged the highest aggression score observed in an arena across all replicates of a nest-pairing. Scores of 2 are difficult to unambiguously categorize as aggressive or non-aggressive and were excluded from the average aggression calculation (see Aggression Assay Methods).

**Table 1. *N. fulva* intraspecific aggression scale.**

| Score | Description of Behavior |
|---|---|
| 0 | Ants come in contact and part. |
| 1 | One or both ants antennate the other for 3 or more seconds OR trophallaxis OR allogrooming |
| 2 | One ant bites and then releases the other. |
| 3 | One ant bites and holds on for more than 3 seconds OR bites and releases two or more times consecutively. |
| 4 | One ant bites and holds the other while curling its gaster tip under its body till the gaster is parallel with substrate, acidopore pointing at its opponent. Aggressor does not spray or otherwise release formic acid. |
| 5 | One ant sprays formic acid at its opponent or both ants actively fight, typically forming a ball. |

Scale used for scoring interactions between workers in 5 x 5 neutral arena interaction assays. The most aggressive behavior observed in a 20 second scan of the arena was scored. Scores of 3 or greater represent increasing levels of aggression. Results provide more detail on behaviors observed in interactions. Behaviors can be seen in supporting information (S1 Video).

## Aggression assays and colony membership

We tested whether non-aggressive nest-pairs and the networks they form meet the expectations for colony membership. If aggression assay results reliably signal colony membership, non-aggressive nests should form spatial clusters without aggressive nests intermixed. We used logistic regression on separation distances and mapping of colony network assignments to evaluate this property. Further, non-aggressive nests should also be part of networks of mutually non-aggressive interactions comprised of 'transitive, non-aggressive' subnetworks: 3-way, nest-pair networks in which nest A workers are non-aggressive to nest B workers which are non-aggressive to nest C workers which are non-aggressive to nest A workers. These subnetworks should also contain few or no 'intransitive, non-aggressive' subnetworks: nest A workers are non-aggressive to nest B workers which are non-aggressive to nest C workers, but nest C workers are aggressive to nest A workers. To assess whether aggression in the neutral arena context creates interaction networks consistent with the expectations for a marker of colony membership, we used the observed frequency of non-aggressive and aggressive nest-pair interactions in the native-range data set to calculate the null probability of observing either 3-way, 'transitive, non-aggressive' nest-pair networks or 3-way, 'intransitive, non-aggressive' nest-pair networks by chance. The null expectation for a given type of 3-way interaction network was calculated by multiplying the observed frequencies of its component interactions. For example, the null expectation for the frequency of 'transitive, non-aggressive' 3-way subnetworks was the frequency of non-aggressive nest-pairs cubed. We then compared the observed frequencies of these types of 3-way, nest-pair networks to these null expectations using Likelihood Ratio $X^2$ tests.

Based on the results of the separation distance and transitivity analyses described above (see Results), we assigned nests in the intensively sampled sites in the native range a colony identity derived from the extended network of non-aggressive interactions in which that nest participated. To be conservative in assigning nests to nest-networks (colonies), all nest-pairs for which the average maximum aggression was one or less were defined as non-aggressive and interpreted as interactions between distinct nests belonging to the same colony, while interaction score averages that were three or greater were defined as aggressive and arising from interactions between members of distinct colonies. All nest-pairs with average scores greater than 1 but less than 3 were excluded. The latter were uncommon, applying to 21 of 155 native range nest-pairings.

To compare the frequency of non-aggressive interactions between sites and regions in the native range, we combined the Uruguay roadside survey data and the data from the more intensively sampled sites in Argentina into a single analysis. Both regions had data for nest-pair separation distances up to 1400 m with similar frequency distributions. However, many fewer nest-pairs with separation distances greater than 1400 m were sampled in Argentina. So, for this analysis, we excluded all nest-pairs separated by more than 1400 m. However, including all nest-pairs does not alter the differences reported herein.

Statistical analyses were performed with JMP$^{©}$ statistical software [36] and maps were generated using ArcGIS mapping software [37]. All work adhered to relevant permitting regulations. Field work was conducted on public, and private lands. The Administración de Parques

Nacionales de Argentina, the Dirección de Recursos Naturales in the Ministerio de Turismo de la Provincia de Corrientes, Argentina, and the Secretaria de Ambiente y Desarrollo Sustentable de la Nación provided permits to collect, transport, and export ant specimens from Argentina. The Ministerio de Vivienda Ordenamiento Territorial y Medio Ambiente, Dirección Nacional de Medio Ambiente División Fauna de Uruguay provided permits to collect and export specimens from Uruguay. The United States Fish and Wildlife Service, Travis County, the City of Austin, and various private landowners provided permission to collect ants on their lands. No protected species were sampled in this work.

## Results

### Queen numbers

In the native range, 115 *N. fulva* nests were collected. Queens were encountered in 30 of these nests with a median of 1 (1–3.3) queens per nest (median (interquartile range)), 13 contained multiple queens with a maximum of 33 queens encountered in a single nest. In the introduced range, 65 *N. fulva* nests were collected. Queens were encountered in 52 of these nests with a median of 2 (2–15) queens per nest, 45 nests contained multiple queens with a maximum of 112 queens encountered in a single nest. Sites in the native range differed in the number of queens found per nest (Wilcoxon: $X^2 = 10.6$, DF = 3, $N = 115$, $P < 0.02$) with more queens per nest found in the Uruguay roadside survey (1.23±2.70) than in Argentina Site 2 (0.05±0.23) or Argentina Site 3 (0±0), while Argentina Site 1 (the town site) was intermediate (1.18±5.2).

### Characterizing aggression: Native range

In the 5x5 neutral arena assays, aggressive interactions between workers from distinct nests were commonly observed in the native range (Fig 2). These behaviors appeared highly stereotyped. Throughout an interaction, one or more ants would exhibit aggressive behaviors towards a single focal ant. The ant that was the focus of aggression (aggressee) typically remained immobile. These interactions were sometimes protracted, lasting beyond the 6-minute arena observation interval. Aggression levels in nest-pairs where aggressive behaviors were observed (scores of 3 or greater) escalated to the highest level of aggression (balling and fighting or spraying of formic acid) in only 15 out of 63 aggressive nest-pairings (24%). Whereas among this same set of aggressive nest-pairs, aggression commonly escalated to the point of the aggressor biting and holding its opponent while pointing its acidopore at the aggressee without spraying formic acid (category 4) (63% of aggressive nest-pairings) (Fig 2). That formic acid was not used was inferred by the lack of any reaction by the aggressee ant. Mortality arising from aggression was rarely observed and ants engaged in aggressive interactions that did not escalate to level 5 appeared uninjured at the end of the observation interval. Some arenas were set aside to observe the longer-term outcome of aggressive interactions. These trials typically ended with two groups of ants clustered in separate areas on the bottom of the vial. In aggressive interactions that did not escalate to balling and fighting or acid spraying, after protracted periods of one-sided aggressive behavior, the aggressor was sometimes observed carrying the immobile aggressee up the side of the vial. The ants being carried were in an immobile, curled posture but were not dead. These aggressive behaviors were only observed in the native range.

### Intraspecific aggression: Introduced range

Across 126 introduced-range interaction assays only a single aggressive interaction (score of 3) was observed. There was no difference between the average maximum interaction scores for

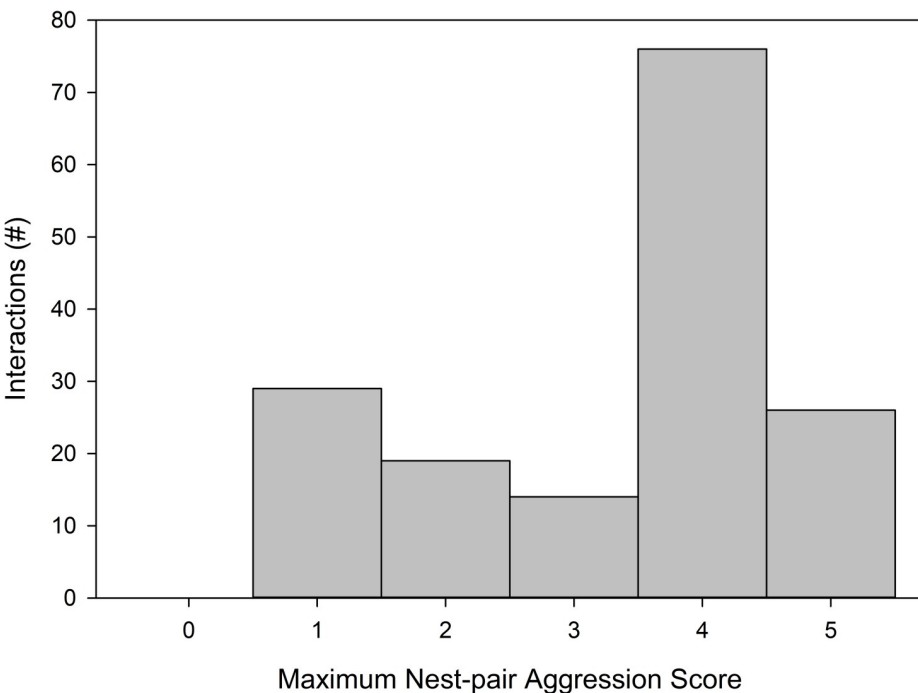

**Fig 2. Distribution of aggression scores observed for all nest-pair interactions in the native range.** Data plotted are the maximum observed aggression score across all replicate arenas. Scores of 3 or greater are aggressive while scores of 1 or lower arise from colony-mate interactions. Aggressive interactions typically plateau at level 4: gaster bending, a ritualized behavior resembling formic acid spraying (see S1 Video).

'nestmate', 'close', 'distant' or 'different region' nest-pairs (Wilcoxon: $X^2 = 5.8$, $N = 63$, $P = 0.12$) (Fig 3A). When comparing nestmate interactions to all nest-pairings comprised of nests from distant local-populations, same nest-pairings had marginally lower interaction scores due to a lower frequency of non-aggressive, category 1 interactions (prolonged antennation, trophallaxis, or allogrooming) (Wilcoxon: $X^2 = 3.5$, $N = 52$, $P = 0.06$). Thus, aggression was absent between workers from nests collected across an area spanning much of the introduced range.

## Intraspecific aggression: Native range

In the native range, inter-nest aggressive interactions were common and interaction scores depended upon nest separation, with 'close' nest-pairs exhibiting lower aggression levels compared to 'distant' nests separated by 500 m or more (Wilcoxon: $X^2 = 11.2$, $N = 60$, $P < 0.0009$) (Fig 3B).

If neutral arena assays reveal colony membership, then non-aggressive interactions should be transitive: making 3-way, 'transitive, non-aggressive' subnetworks more common than expected by chance and making 3-way, 'intransitive, non-aggressive' subnetworks less common. Given the observed frequencies of aggressive and non-aggressive interactions for all nest-pairings involved in 3-way interaction networks, 'intransitive, non-aggressive' networks occurred less frequently than the null expectation (26.1%), occurring in 11.0% of all 3-way interaction networks (Likelihood Ratio $X^2$: $X^2 = 10.3$, $N = 73$, $P < 0.002$). Equally 'transitive, non-aggressive', 3-way networks were more common than the null expectation (5.1%), occurring in 11.0% of all 3-way interaction networks (Likelihood Ratio $X^2$: $X^2 = 3.9$, $N = 73$, $P < 0.05$). When we applied a more conservative definition of aggressive and non-aggressive

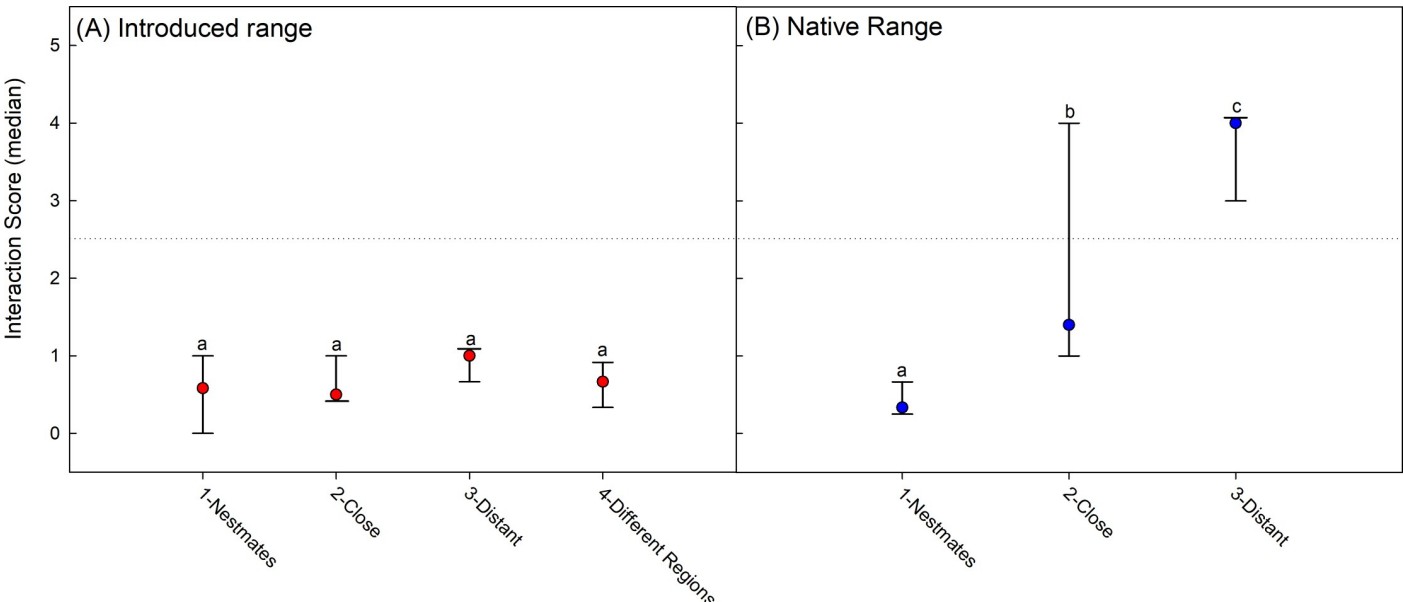

**Fig 3. Introduced—native range contrast ofinteraction scores across nest separation distance categories.** Error bars present the interquartile range of interaction scores. 'Same' nest-pairs–worker ants collected from the same nest, separated and then recombined; 'Close' nest-pairs–ants collected from nests separated by about 100 m; 'Distant' nest-pairs–ants collected from nests from different sites; 'Different-region' nest-pairs–ants collected from nests in different geographic regions (i.e. Texas vs Florida) (A) Introduced range. (B) Native range.

interactions (eliminating 3-way networks that included pairs with average interaction scores near 2 (see methods)), all intransitive, non-aggressive networks were eliminated from the data set. Further, if interaction assays reveal colony membership, aggression should increase with separation distance leading to local neighborhoods of non-aggressive nest networks. At two of the three Argentina sites where nests were intensively sampled, the probability that nests exhibited aggression increased with increasing nest separation distance (Fig 4). Thus, interaction assays reflected the properties expected for colony membership.

Given the previous finding that 3-way subnetworks of non-aggressive interactions are transitive, we assigned nests to colonies based on the extended network of non-aggressive nest-pair interactions in which they participated. Colonies defined on this basis were spatially contiguous, non-overlapping entities, without aggressive nests intermixed (Fig 5). Colonies (networks of nests with mutually non-aggressive workers) were also highly variable in spatial extent. Workers from 46% (29/63) of nests from intensively-sampled, native-range sites were aggressive to all alien nest workers they confronted including the nearest neighbor nests encountered. However, in addition to these small, potentially single-nest colonies, expansive, polydomous colonies were found in all intensively-sampled, native-range sites. Within these expansive colonies, the median distance between non-aggressive nests was 55 (33–103) m (median (interquartile range)) (Fig 6). As no site was completely sampled and *N. fulva* nests may occur in inaccessible locations (e.g. deep in soil), these distances likely underestimate the area occupied by individual colonies. Further supporting the commonplace nature of spatially expansive, polydomous colonies in the native range, in roadside surveys of disturbed, flood prone habitats in Uruguay in which two to three nests were sampled per site, nest-pairs at nine out of 11 sites were non-aggressive. These non-aggressive nest-pairs were typically widely separated: 85 (47–212) m (median (interquartile range)).

The likelihood of encountering non-aggressive nest-pairs (a surrogate for frequency or extent of spatially-expansive colonies) varied with area sampled in the native range. There was

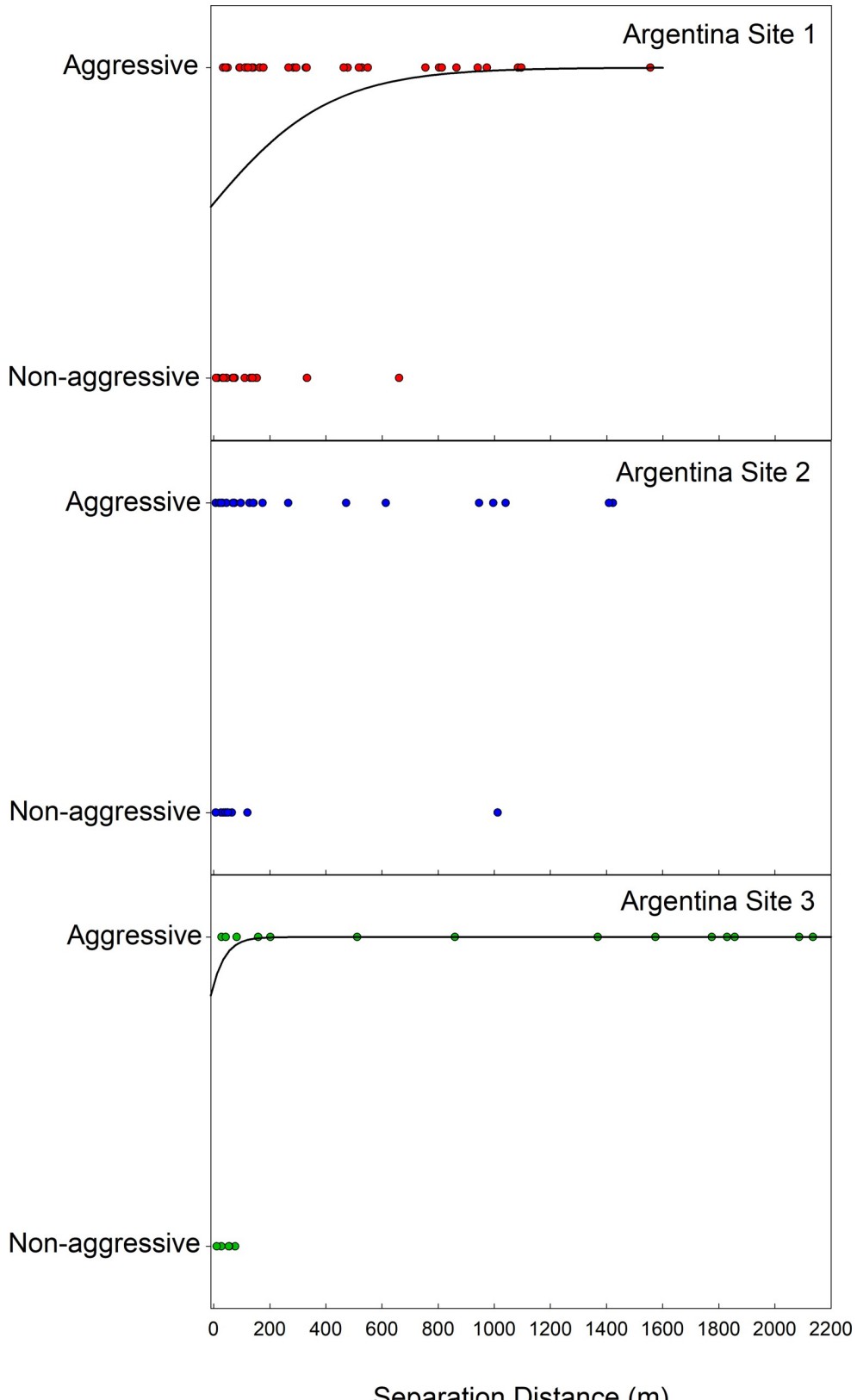

**Fig 4. Likelihood of *N. fulva* nest-pair aggression as a function of distance between nests.** Data are from the three intensively sampled sites in the native range. For sites where separation distance is significantly associated with likelihood of aggression between nests, the lines plot the best logistic regression fit to the data. (A) Argentina Site 1: $N = 52$, OR = 1.004, $P < 0.002$; (B) Argentina Site 2: $N = 32$, $P = 0.18$; (C) Argentina Site 3: $N = 19$, OR = 1.02, $P < 0.0009$.

no difference between the intensively sampled sites in Argentina in overall frequency of non-aggressive interactions (Likelihood Ratio $X^2$: $X^2 = 0.39$, DF = 1, $N = 95$, $P = 0.82$). However, there was a higher likelihood of encountering non-aggressive nest-pairs in the Uruguay road-side survey than in the intensively sampled sites in Argentina treated as a whole (Likelihood Ratio $X^2$: $X^2 = 5.1$, DF = 1, $N = 111$, $P < 0.03$).

## Discussion

In contrast to its introduced range in North America, in northern Argentina and Uruguay *N. fulva* workers from different nests often exhibit intraspecific aggressive behaviors. Aggressive behaviors expressed during intraspecific interactions in *N. fulva* differ markedly from aggression in interspecific conflicts in the same neutral arena context. Interspecific encounters between *N. fulva* and some of its prominent competitors in South America, *Solenopsis invicta* and *Linepithema humile*, typically rapidly escalate to the use of chemical defense compounds and fighting to the death [15,38]. In contrast, aggressive encounters amongst *N. fulva* workers followed a stereotyped series of escalating aggressive behaviors [39]. These interactions typically plateaued at a ritualized aggressive behavior (level 4), in which venom was not deployed and neither ant was injured (Fig 2). Interactions were typically one-sided with an aggressor ant biting an immobile opponent (S1 Video).

The most commonly observed peak interaction score in aggressive nestmate pairings was level 4: one ant bites and holds the other, curls its gaster (abdomen) under its body until the acidopore (venom gland opening at tip) is parallel to the substrate pointing at its opponent but venom is not released. This behavior, termed "gaster bending", constitutes a ritualized aggressive behavior that appears to be a derived from the more typical combat behavior of formic acid spraying. It has been reported in intraspecific territorial interactions in two other Formicine species but only one employs it in worker-worker interactions [40,41]. This ritualized aggression contrasts with other supercolonial ant species in which intraspecific interactions between members of distinct supercolonies are characterized by rapid escalation to fighting to the death [21,42,43], or an absence of aggression at least within social form (eg: polygyne *S. invicta* [44]).

The absence of intraspecific aggression in *N. fulva* is transitive, 'intransitive non-aggressive' networks are rare, and mutually non-aggressive nests form clusters without aggressive nests intermixed (Fig 5). Thus, the results of interaction assays meet the expectations for an indicator of colony identity and provide a useful way to define colony boundaries in the native range of *N. fulva*. However, intransitive subnetworks did occur in the data set. 'Intransitive, non-aggressive" sub-networks all had at least one nest pair link with average interaction scores of intermediate value (between 1 and 3), and all of these intermediate interaction scores arose from replicated sets of arenas with a mix of aggressive and non-aggressive outcomes. In this data set, 'intransitive, non-aggressive' subnetworks appear to be the result of interactions between non-nestmates that inconsistently escalated to aggressive interactions in this neutral arena context. It is unclear if there is an underlying similarity in signal cues between nest pairs that led to inconsistent assay results. However this inconsistency highlights the importance of standardizing assay conditions, replicating assays, minimizing exposure to artificial odor cues,

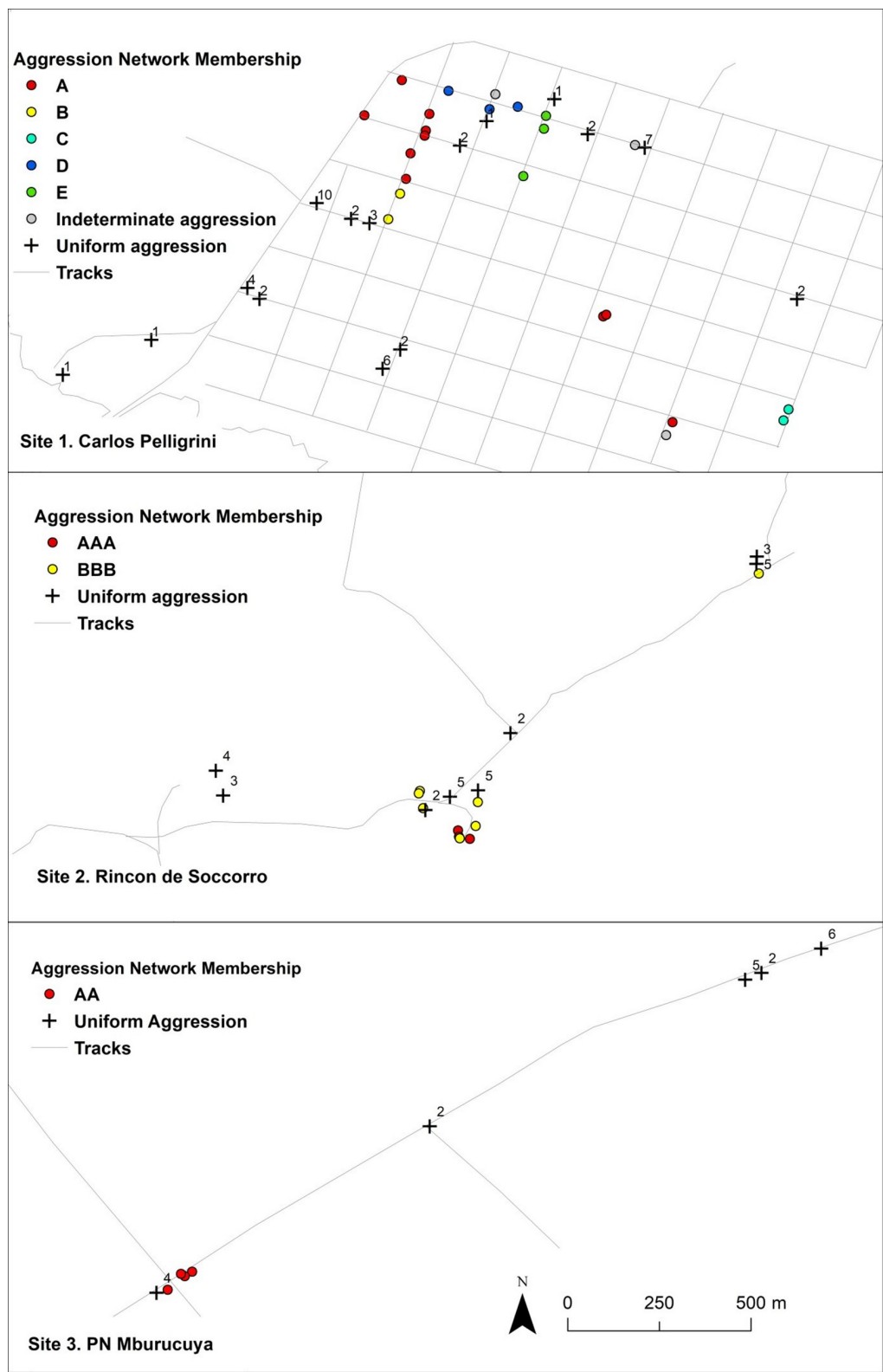

**Fig 5. Maps of non-aggressive nest networks (polydomous colonies) for the 3 intensively sampled sites in the native range (Corrientes Province, Argentina).** Colored dots are nest locations for nests that exhibited non-aggressive interactions with other sampled nests. Dots of the same color are nests that are members of the same nest network as defined by the absence of aggression with other nests in that network. Crosses are locations of nests that were aggressive

to all other nests with which they were confronted, including the nests that were their nearest neighbor. Numbers next to crosses indicate the number of aggressive nests they were paired with. Grey circles are nests that participated only in interactions that escalated to intermediate aggression levels.

and, when feasible, within a set of neutral arena assays, including known aggressive and non-aggressive nest pairs as positive and negative controls for assay conditions.

All intensively sampled sites in the native range were comprised of spatially expansive colonies intermixed with colonies of smaller size. These smaller colonies may be potentially

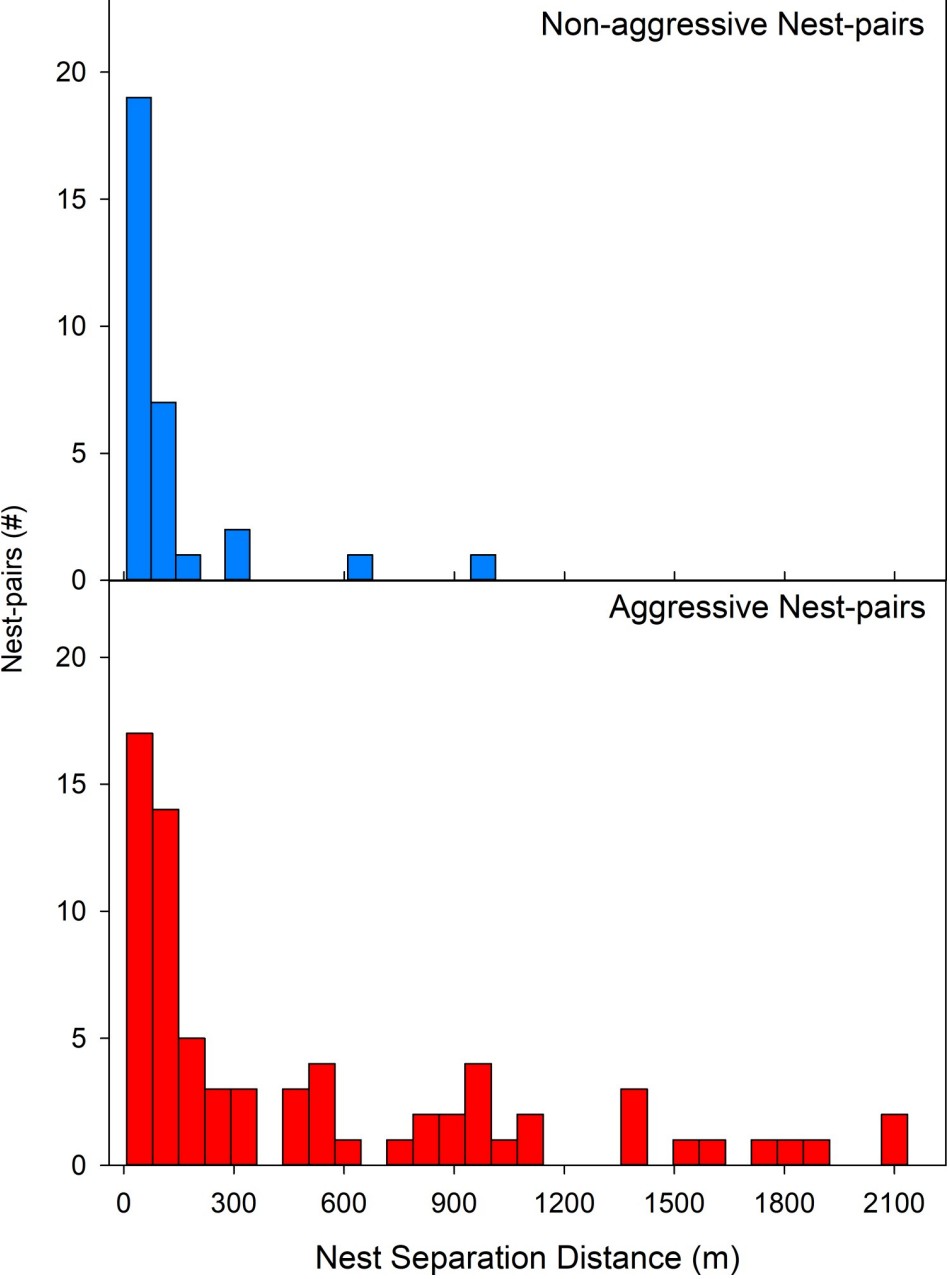

**Fig 6. Distribution of nest separation distances for non-aggressive and aggressive nest-pairs.** Data combines all intensively sampled sites in Argentina.

restricted to a single nest (Fig 5). This contrasts with the absence of intraspecific aggression at any spatial scale within the introduced range. This general pattern of smaller colony sizes in the native range than in the introduced range is shared by other supercolonial invasive species such as *L. humile* [1,45], *W. auropunctata* [21], and *Pheidole megacephala* [46]. In the native range, *N. fulva* colonies appear to vary from possibly monodomous colonies to expansive, polydomous colonies that approach the size of supercolonies. The lower limit of what defines a supercolony being defined as a colony sufficiently large that it is unlikely that individuals from distinct nests will interact [47,48]. It is possible that this colony-size variation documented in *N. fulva* may be underpinned by a polymorphism in social organization, analogous to the monogyne vs polygyne dichotomy in fire ants [49,50]. Similar variation in social organization within their respective native ranges has been suggested in both *W. auropunctata* [21] and *L. humile* [45].

Non-aggressive nest-pairs were more common in the roadside-survey in Uruguay than in the intensively sampled sites in Argentina. This observation implies that the relative frequency and likely the spatial extent of colonies also varies in a site-specific manner. In concert, the number of queens per nest was also higher in the road-side Uruguay survey than in the intensively sampled sites in Argentina. One possible driver of this variation is disturbance. Uruguay roadside sampling sites were in general subject to high levels of anthropogenic disturbance. Further, most were adjacent to water courses and thus additionally subject to disturbance due to flooding. Supercolonial ants may be able to survive disturbance events more successfully or grow more rapidly post-disturbance as colonies fragmented by flooding or other habitat disruption are more likely to be reproductively competent and less likely to suffer the costs of intraspecific competition as they recover. In another supercolonial invasive ant species (*Wasmannia auropunctata*), disturbance has been found to be associated with expansive colonies and dense local-populations [20,51]. Another, non-mutually exclusive possibility is that colony fragments are translocated by people in the native range as they commonly are in the introduced range. If these human-distributed colonies are somehow more suited to human modified habitats and proliferate after introduction, then a restricted subset of colonies may become widespread in areas of greater human activity. Further, this may constitute a selective process in which human habitat modifications and colony translocations in the native range may have selected for traits, like supercoloniality, that facilitate the ability of *N. fulva* to invade outside of their native range [52]. Genetic analyses of spatially-expansive colonies in periurban and remote settings with variable flooding regimes would provide interesting insight into these hypotheses.

In the introduced range, ants from sites in distinct geographic regions separated by as much as 1600 km were no more aggressive to one-another than were workers from the same nest. This equivalence, combined with the manifestation of intraspecific aggression in native local-populations under the same conditions, indicates that disjunct local-populations of *N. fulva* within North America share the same supercolony identity. The shared supercolony identity of *N. fulva* in North America is also supported by the limited, and high overlapping genetic diversity of nests from disjunct local-populations [15]. As no intraspecific aggression has been found to date within or between local-populations in North America [see also 15], the North American *N. fulva* population appears to be unicolonial, comprised of many disjunct, mutually-tolerant, local-populations belonging to the same supercolony. Given the frequency of aggression in the native range, this unicolonial state is an emergent property of introduction into a novel environment. As in other invasive ant systems [53,54], distinct supercolonies may eventually be found in North America, however the problem of understanding the origin of extremely large supercolonies will remain.

Two categories of hypotheses have been put forward, primarily in efforts to understand Argentine ant populations, to explain the post-introduction emergence in invasive ants of

unicoloniality or large-scale supercoloniality. The first category posits, via distinct mechanisms, that these very large societies result from a reduction in genetic diversity post introduction that allows for the fusion of nests from previously distinct colonies [35,54]. In these cases, unicoloniality, or extremely large supercolonies, represents an innovation arising post introduction derived from changes in gene frequencies. The second category of hypothesis posits that unicoloniality arises from a developmental transition. Individual colonies in the native range possess the potential to achieve unicoloniality but are constrained by ecological factors. In the introduced range, these ecological factors that limit colony expansion (e.g. intraspecific competition, parasites, pathogens, potentially more effective competitors) are absent allowing single colonies to expand with few limits [47,55]. Herein, we demonstrate the existence of polydomous, polygynous, spatially-expansive colonies in the native range of *N. fulva*. Thus, the simpler explanation for the emergence of unicoloniality, unlimited expansion of a single colony, appears viable as these expansive, native-range colonies resemble local-populations of the North American supercolony in all physical characters aside from spatial scale. However, we also observe that the aggressive behaviors that enforce colony boundaries appear ritualized and tend to not escalate to fighting to the death. Perhaps this results in a lower barrier to the fusion of nests from distinct colonies. Current genetic data appears consistent with either route to unicoloniality; nestmate workers from the introduced range exhibit lower relatedness than those from the native range, but relatedness among introduced-range nestmates is considerably greater than zero when viewed in a global frame of reference [15]. Ultimately, differentiating these divergent routes to unicoloniality requires contrasting the relatedness and genetic diversity among nests within the spatially-expansive, native-range colonies documented herein with these same quantities within local-populations of *N. fulva* in North America.

## Conclusions

Intraspecific aggression in *N. fulva* is ritualized and its expression enforces colony boundaries. However, within the introduced range, intraspecific aggression is absent at all spatial scales tested, supporting the inference that this ant is unicolonial within North America. Intraspecific aggression is common in the native range, and colonies vary widely in spatial extent. On the upper end of this variation exist multiqueen colonies distributed amongst many nests that span hundreds of meters; entities that approach the lower limit of what could be described as supercolonies. Unicoloniality in the North American population is thus very likely a state derived from these large societies in the native range. Therefore, understanding the mechanisms that gave rise to introduced range unicoloniality requires a careful description of the behavior and genetics of these large, native range colonies.

## Supporting information

**S1 File. Intraspecific interaction data, tawny crazy ants.**
(XLSX)

**S1 Table. Locality information and sampling intensity for all study sites.**
(DOCX)

**S1 Video. Intraspecific aggressive behaviors in tawny crazy ants.** Video displays behaviors expressed in intraspecific interactions in tawny crazy ants (*Nylanderia fulva*) that are described in Table 1. Filmed in Nuevo Berlin, Uruguay, April 2016.
(MP4)

## Acknowledgments

We thank Cecilia Cabrera, Carolina Gomila, and Brett Morgan for technical work. Luis Calca-terra provided advice on field sites in Argentina. David Oi provided ants from Florida for interaction assay tests. The Estancia Rincón del Socorro and The Conservation Land Trust Argentina provided us with access to their property and Sebastián Di Martino and Ignacio Jimenez provided advice and logistical support. David Holway and Ed Vargo provided useful critique of the manuscript.

## Author Contributions

**Conceptualization:** Edward G. LeBrun, Robert M. Plowes, Lawrence E. Gilbert.

**Data curation:** Edward G. LeBrun.

**Formal analysis:** Edward G. LeBrun.

**Funding acquisition:** Edward G. LeBrun, Robert M. Plowes, Lawrence E. Gilbert.

**Investigation:** Edward G. LeBrun, Robert M. Plowes.

**Methodology:** Edward G. LeBrun, Robert M. Plowes.

**Project administration:** Edward G. LeBrun, Patricia J. Folgarait, Martin Bollazzi.

**Resources:** Edward G. LeBrun, Patricia J. Folgarait, Martin Bollazzi.

**Supervision:** Edward G. LeBrun, Patricia J. Folgarait.

**Validation:** Edward G. LeBrun.

**Visualization:** Edward G. LeBrun, Robert M. Plowes.

**Writing – original draft:** Edward G. LeBrun.

**Writing – review & editing:** Robert M. Plowes, Patricia J. Folgarait, Lawrence E. Gilbert.

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
