## [Decision Letter · Decision Letter 0]

29 Oct 2019

PONE-D-19-25167

Ritualized aggressive behavior reveals distinct social structures in native and introduced range tawny crazy ants

PLOS ONE

Dear Dr LeBrun,

Thank you for submitting your manuscript to PLOS ONE. After careful consideration, we feel that it has merit but does not fully meet PLOS ONE’s publication criteria as it currently stands. Therefore, we invite you to submit a revised version of the manuscript that addresses the points raised during the review process.

Please consider the corrections and improvement suggested

We would appreciate receiving your revised manuscript by Dec 13 2019 11:59PM. To enhance the reproducibility of your results, we recommend that if applicable you deposit your laboratory protocols in protocols.io, where a protocol can be assigned its own identifier (DOI) such that it can be cited independently in the future. For instructions see: http://journals.plos.org/plosone/s/submission-guidelines#loc-laboratory-protocols

We look forward to receiving your revised manuscript.

Kind regards,

Nicolas Chaline

Academic Editor

PLOS ONE

Journal Requirements:

2.  We note that Figure [1] in your submission contains a map image which may be copyrighted. All PLOS content is published under the Creative Commons Attribution License (CC BY 4.0), which means that the manuscript, images, and Supporting Information files will be freely available online, and any third party is permitted to access, download, copy, distribute, and use these materials in any way, even commercially, with proper attribution. For these reasons, we cannot publish previously copyrighted maps or satellite images created using proprietary data, such as Google software (Google Maps, Street View, and Earth). For more information, see our copyright guidelines: http://journals.plos.org/plosone/s/licenses-and-copyright.

1.    You may seek permission from the original copyright holder of Figure(s) [1] to publish the content specifically under the CC BY 4.0 license. 

Reviewers' comments:

Reviewer's Responses to Questions

**Comments to the Author**

1. Is the manuscript technically sound, and do the data support the conclusions?

Reviewer #1: Yes

2. Has the statistical analysis been performed appropriately and rigorously? 

Reviewer #1: Yes

3. Have the authors made all data underlying the findings in their manuscript fully available?

Reviewer #1: Yes

4. Is the manuscript presented in an intelligible fashion and written in standard English?

Reviewer #1: Yes

5. Review Comments to the Author

Reviewer #1: This paper provides new data and novel insights into the biology of an invasive ant by providing data on its colony structure in both its native and introduced range. The paper is very well written. The introduction does a very good job putting this work into a general context and the methods are clear allowing the study to be replicated.

All of my suggestions are minor.

Editorial Suggestions:

Line 85-86. Change to “However, few of these species have had their social organization described in both native and introduced populations.”

Line 177. The comma after “Texas” can be deleted.

Line 121. Change “Not collected” to “missed by this collection method.”?

Line 208. The first sentence of this section (Aggression scale) is a bit of confusing without more context. “Observations of nest-pairs not included in further analyses were used to adapt standard aggression scales developed to quantify interactions between Argentine ants[34,35] to interactions between N. fulva workers” Can you better explain the context of what you mean / what you are doing here.

Line 401. “Stark” can be deleted.

Line 404-408. This is interesting and worth following up on. The authors could consider adding a graph / table comparing intra- versus inter-specific pattern of aggression to quantify this difference as part of the paper (just a suggestion).

Line 424. This paragraph could be better developed in terms of social structure generally in ants. Are there other examples that can be discussed that similar to what is seen here? Are there examples of non-transitive behaviors among colonies. Why do these occur?

6. PLOS authors have the option to publish the peer review history of their article (what does this mean?). If published, this will include your full peer review and any attached files.

Reviewer #1: No

---

## [Author Response · Author response to Decision Letter 0]

5 Nov 2019

Dear PLOS Editorial Staff,

Please find below my point-by-point response to comments of the academic editor and reviewer. Many thanks for your efforts on this manuscript.

“To enhance the reproducibility of your results, we recommend that if applicable you deposit your laboratory protocols in protocols.io, where a protocol can be assigned its own identifier (DOI) such that it can be cited independently in the future.”

I have uploaded a detailed protocol for how to conduct aggression assays. Protocol DOI: dx.doi.org/10.17504/protocols.io.8z9hx96.

“Journal Requirements:

 Please ensure that your manuscript meets PLOS ONE's style requirements, including those for file naming.”

I have reviewed the style templates and made all necessary changes in the document to match. 

"2. We note that Figure [1] in your submission contains a map image which may be copyrighted. All PLOS content is published under the Creative Commons Attribution License (CC BY 4.0), which means that the manuscript, images, and Supporting Information files will be freely available online, and any third party is permitted to access, download, copy, distribute, and use these materials in any way, even commercially, with proper attribution. For these reasons, we cannot publish previously copyrighted maps or satellite images created using proprietary data, such as Google software (Google Maps, Street View, and Earth). For more information, see our copyright guidelines: http://journals.plos.org/plosone/s/licenses-and-copyright."

This is not a copyrighted image. This map image was generated in Arc GIS using our data for the purpose of this publication. It is fine if this is published under the Creative Commons Attribution License. To clarify this, I have added a statement to the methods citing the software used to generate the map.

The two academic editor comments following this one relate to the interpretation of this figure as copyrighted material. It is not, so, for brevity, these comments have not been copied into this response.

Reviewers' comments:

Reviewer's Responses to Questions

Comments to the Author

1. Is the manuscript technically sound, and do the data support the conclusions?

Reviewer #1: Yes

2. Has the statistical analysis been performed appropriately and rigorously? 

Reviewer #1: Yes

3. Have the authors made all data underlying the findings in their manuscript fully available?

Reviewer #1: Yes

4. Is the manuscript presented in an intelligible fashion and written in standard English?

Reviewer #1: Yes

5. Review Comments to the Author

Reviewer #1: This paper provides new data and novel insights into the biology of an invasive ant by providing data on its colony structure in both its native and introduced range. The paper is very well written. The introduction does a very good job putting this work into a general context and the methods are clear allowing the study to be replicated.

Thank you.

All of my suggestions are minor.

Editorial Suggestions:

Line 85-86. Change to “However, few of these species have had their social organization described in both native and introduced populations.”

Done

Line 177. The comma after “Texas” can be deleted.

Done.

Line 121. Change “Not collected” to “missed by this collection method.”?

Done.

Line 208. The first sentence of this section (Aggression scale) is a bit of confusing without more context. “Observations of nest-pairs not included in further analyses were used to adapt standard aggression scales developed to quantify interactions between Argentine ants[34,35] to interactions between N. fulva workers” Can you better explain the context of what you mean / what you are doing here.

I have clarified this section. It now reads, “Standard aggression scales have been developed to quantify intraspecific interactions in Argentine ants (Linepithema humile) [34,35]. We conducted preliminary interaction assays to adapt this scale to incorporate the behaviors observed in interactions between N. fulva workers. Data from these preliminary assays are not included in the analyses presented herein.”

Line 401. “Stark” can be deleted.

Done.

Line 404-408. This is interesting and worth following up on. The authors could consider adding a graph / table comparing intra- versus inter-specific pattern of aggression to quantify this difference as part of the paper (just a suggestion).

We agree that this is interesting. However, the data for interspecific aggression were collected for a paper on chemical ecology prior to observing intraspecific interactions in the native range and thus prior to the development of this behavioral scale. The qualitative difference is obvious as the interspecific assay was predicated on the ants deploying their chemical defenses rapidly which they did in all interactions (and almost never do in intraspecific interactions). However, these interactions were not scored with the same behavioral scale and thus a direct quantitative comparison is not possible without additional data collection. As this is a related but ancillary issue to the topic of the paper, we are not undertaking this at this time.

Line 424. This paragraph could be better developed in terms of social structure generally in ants. Are there other examples that can be discussed that similar to what is seen here? Are there examples of non-transitive behaviors among colonies. Why do these occur?

 I have added an analysis of why these interactions occur and my suggestions for their significance for studies of ant social structure. 

I have done so. I saw no way to approve or submit the uploaded figures, so I am assuming that you have access to what I uploaded. I reviewed the figures and they all looked good to me. It was unclear whether you also wanted Tables uploaded in this format, so I went ahead and did so.

I thank you and the Reviewer for your efforts.

Sincerely,

Edward LeBrun

---

## [Editor Report · Decision Letter 1]

8 Nov 2019

Ritualized aggressive behavior reveals distinct social structures in native and introduced range tawny crazy ants

PONE-D-19-25167R1

Dear Dr. LeBrun,

We are pleased to inform you that your manuscript has been judged scientifically suitable for publication and will be formally accepted for publication once it complies with all outstanding technical requirements.

With kind regards,

Nicolas Chaline

Academic Editor

PLOS ONE
---

## [Editor Report · Acceptance letter]

15 Nov 2019

PONE-D-19-25167R1 

Ritualized aggressive behavior reveals distinct social structures in native and introduced range tawny crazy ants 

Dear Dr. LeBrun:

I am pleased to inform you that your manuscript has been deemed suitable for publication in PLOS ONE. Congratulations! Your manuscript is now with our production department. 

With kind regards,

on behalf of

Professor Nicolas Chaline 

Academic Editor

PLOS ONE